# Prediction of Growth Characteristics and Migration Period of *Spodoptera frugiperda* (Lepidoptera: Noctuidae) According to Temperature

**DOI:** 10.3390/insects13100897

**Published:** 2022-10-02

**Authors:** Seongkyun Lee, Younguk Park, Chohee Hwang, Areum Park, Seokho Lee, Juhyoung Kim

**Affiliations:** Chungbuk Agricultural Research and Extension Services, 46 Gagok-gil, Ochang-eup, Cheongju 28130, Korea

**Keywords:** *Spodoptera frugiperda*, growth characteristics, corn leaf feeding amount

## Abstract

**Simple Summary:**

*Spodoptera frugiperda* is a pest with a wide host range that causes severe damage to the agricultural sector around the world. The species, which was first recorded in South Korea in 2019, has been affecting corn farmers by harming the leaves and fruits. Unfortunately, measures to initially respond to the species have not been implemented appropriately due to the lack of information on the growth characteristics of *S*. *frugiperda* in the Korean climate. Against this backdrop, we analyzed the growth characteristics and degree of damage to corn leaves at different temperatures and found that the optimal temperature for the growth of *S*. *frugiperda* was in the range 28–32 °C, which is the temperature range during summer in Korea. It is most active during this season, affecting more number of farmers progressively. In this regard, more efforts are needed to control the pest in summer in Korea.

**Abstract:**

In 2019, an outbreak of *Spodoptera frugiperda* (Smith) was first reported in Korea. This study aimed to determine the growth rate and feeding amount of *S. frugiperda* by temperature to establish the right time window for its control and management. Linear regression analysis was used to determine the growth period and thermal requirements of *S. frugiperda*. The longest growth period of 97.2 ± 1.2 days was observed at 16 °C, and the shortest growth period of 15.5 ± 0.7 days was observed at 36 °C. In terms of each growth stage, the pupal period was the longest at all temperatures, followed by the egg period. The maximum corn leaf feeding amount (6.61 g) was observed for the larvae grown at 16 °C, and the minimum (2.9 g) was observed at 36 °C. However, the daily feeding amount of *S. frugiperda* larvae was the highest at 28 °C and 32 °C. The hatching rate according to temperature exceeded 70% at 24 °C, 28 °C, and 32 °C, and the survival rate of larvae and pupae was 100% at 24 °C to 32 °C. Based on these results, a temperature range of 28 °C to 32 °C is proposed as the optimum temperature for the growth of *S. frugiperda*.

## 1. Introduction

*Spodoptera frugiperda* Smith (Lepidoptera: Noctuidae) is a lepidopteran pest that infests more than 80 crops, including corn, rice, sorghum, sugarcane, cotton, and other vegetable crops [1]. The larvae of *S. frugiperda*, in particular, have caused severe damage to monoculture crops such as corn, sorghum, cotton, and soybean [2,3,4], and if infesting the early stages of corn growth, they can decrease the yield by up to 70% [5,6]. In addition, a single larva can attack multiple hosts during development [7]. According to *S. frugiperda* population studies, the same strains attack cotton and corn, and can infect crops while migrating between corn and cotton fields [8,9,10,11]. Additionally, despite having a short lifespan, the adult female is extremely fertile, laying approximately 1000 eggs per individual and depositing 100 to 300 of them on the leaf [12]. The first and second instar larvae of *S. frugiperda* only inflict damage on the corn epidermis, leaving traces of damage in the form of glass windows [13].

In general, the timing of applying pest control against a specific pest at an agricultural site is based on the occurrence of the pest. However, there is no information on the occurrence of *S. frugiperda* in Korea, since it has been only recently confirmed that they occur in Korea [14]. Therefore, there is a high chance of missing the timing for pest control at the agricultural site.

*S. frugiperda* is native to the tropical and subtropical regions of the Americas. Until 2015, this species was considered endemic to the Americas. However, it was first reported in Africa when it caused severe damage to corn crops in several countries in 2016 [15]. In Asia, it was first reported in India in July 2018, and rapidly spread to Bangladesh, China, Laos, Myanmar, Sri Lanka, Thailand, and Vietnam [16]. In Korea, the first instance of second and third instar larvae infesting crops in corn fields on Jeju Island was reported in 2019 [14].

Lee et al. (2020) [14] identified two population groups, hap-1 and hap-2, through phylogenetic analysis using the *COI* gene for *S. frugiperda* in Korea and reported that the damage spread every year. It is difficult to precisely identify the degree of damage according to temperature. Therefore, in light of the recent outbreak of *S. frugiperda* in Korea, further studies are required to develop a suitable control model for Korea.

Thus, *S. frugiperda* is an economically significant and hazardous pest in agriculture. A particular problem for Korea is that the wind transports new pests from South China [14], causing the optimal pest-control time to change every year. This makes it difficult to develop an effective first response to control the pests. A growth model that can predict the damage period after adult *S. frugiperda* is introduced by wind is therefore needed. To determine whether to control *S. frugiperda* larvae, there is also a need to determine the relationship between the temperature at the time of damage prediction and the feeding amount of *S. frugiperda* larvae. Thus, this research aimed to define the growth model for *S. frugiperda* at each stage, the occurrence of adults according to period, and the amount of corn leaf feeding according to the temperature.

## 2. Materials and Methods

### 2.1. Sample Collection

The third to fifth instar larvae samples of *S. frugiperda* were collected from corn fields in Maro-myeon, Boeun-gun, Chuncheongbuk-do (36°25′51″ N; 127°49′01″ E) and Geunheung-myeon, Taean-gun, Chungcheongnam-do (36°43′38″ N; 126°14′47″ E). For the experiments, the insects were reared till the F4 generation in an insect breeding room at the laboratory under the conditions of 25 °C temperature, 65% relative humidity, and 16L:8D photoperiod.

### 2.2. Investigation of S. frugiperda Growth According to Temperature

The experiment was conducted in an incubator (HB-302S-2H, Han Baek Scientific Co. in Korea) with the same growth conditions at a temperature of 16 ± 1 °C, 20 ± 1 °C, 24 ± 1 °C, 28 ± 1 °C, 32 ± 1 °C, and 36 ± 1 °C (six conditions), humidity 70% ± 5%, and a photoperiod of 16L:8D. The growth stage of the larvae was determined by the presence or absence of the exuviae of the head. The growth rate of 30 larvae was investigated every 24 h through continuous breeding. Each larva was separated and individually bred in a cylindrical plastic dish with a diameter of 100 mm and height of 40 mm. However, the linear regression analysis of the growth rate at 16 °C and 36 °C was excluded owing to the low survival rate (Appendix A). In addition, to verify the accuracy of the linear regression analysis results, eggs were attached to a leaf of corn planted in a small pot in Cheongju and Boeun-gun in Chungcheongbuk-do and Taean-gun, Chungcheongnam-do in South Korea, and a temperature recorder (TR-72wf) was installed with it. The temperature was recorded until the adults appeared; thereafter, comparisons were made using the linear regression model. The hatching rate of the eggs by temperature was measured based on the number of larvae hatched from 100 eggs. The survival rate of the larvae by temperature was measured based on 100 first larvae growing into adults. The experiment was performed in triplicate.

### 2.3. Investigation of S. frugiperda Corn Feeding Amount According to Temperature

The basic breeding conditions and investigation intervals were identical to those described in Section 2.2. The feeding amount of *S. frugiperda* was measured using a scale capable of measuring up to 0.0001 g (ME204, Mettler Toledo); measurements were rounded up from the fourth decimal place to the third decimal place. The feeding amount was calculated by subtracting the weight of the corn leaf after feeding from that before feeding. However, to exclude the weight loss owing to evaporation from the corn leaf, the moisture reduction rate per corn leaf was calculated in five untreated groups and subtracted from the measured weight. Thus, the calculation method is as follows: Weight before feeding × [(Weight before feeding × (Weight after evaporation of water without treatment/Weight before evaporation of water without treatment)] − Weight after feeding. 

The corn hybrid “Mibaek 2” was cultivated indoors and provided daily as the food source for the insects.

### 2.4. Seasonal Occurrence Trends of Adult S. frugiperda

To investigate the occurrence of adult *S. frugiperda*, pheromone traps were installed in a total of 54 corn fields, nine in each of the following six regions: Cheongju, Goesan, and Boeun of Chungchungbuk-do, Dangjin, Hongseong, and Taean of Chungcheongnam-do. The investigation was conducted at 7-day intervals from April to the end of the outbreak. (Z)-9-Tetradecenyl acetate and (Z)-7-dodecenyl acetate were used as pheromones in a ratio of 100:1. The pheromone lure was replaced every 15 days. Traps, lures, and pheromones were purchased from Green Agrotech in Korea.

### 2.5. Data Analysis

One-way ANOVA was used to analyze the change in *S. frugiperda* growth period according to temperature. The temperatures were 20 °C, 24 °C, 28 °C, and 32 °C and the growth stages were analyzed using 30 units each for eggs, larvae (first – sixth instar), and pupae. The growth rate of *S. frugiperda* was calculated using linear regression analysis [17]. In the functional expression y = ax + b of the linear regression model, y is 1/day, x is the temperature, a is the slope, and the lower temperature threshold is an x value where y becomes 0 and was calculated using −b/a in the functional expression. The calculation of degree-day (DD) is generally DD = (Daily max + min temperature)/2 − threshold temperature; however, in this study, the formula DD = Temperature − threshold was used because *S. frugiperda* breeding was carried out under the same temperature condition. Tukey’s HSD test was used to compare the average *S. frugiperda* growth period and its corn leaf feeding amount. IBM SPSS 25 statistics software was used for the statistical analyses.

## 3. Results

Table 1 presents the growth period for each developmental stage of *S. frugiperda*. Within the test temperature range of 16–36 °C, the longest growth period of 97.2 ± 1.2 days was observed at 16 °C and the shortest growth period of 15.5 ± 0.7 days was observed at 36 °C. In terms of the growth stage, the pupa period was the longest at all temperatures, followed by the egg period. In the case of larvae, the sixth instar was the longest, and the second instar was the shortest. 

The hatch rate of eggs was 84.0% at 24 °C, 85.7% at 28 °C, and 77.3% at 32 °C (Appendix A). The temperatures 24 °C, 28 °C, and 32 °C were most suitable temperatures for hatching, with a hatch rate exceeding 70%. Moreover, the hatching rate exceeded 50% at other temperatures, indicating that the hatching rate is not low as it is protected from external abnormal weather at the egg stage. The survival rates of larvae were 88.0% at 20 °C, 97.3% at 24 °C, 99.0% at 28 °C and 98.7 at 32 °C. However, the survival rate was significantly reduced to 31.0% and 45.3% at 16 °C and 36 °C, respectively, which can be attributed to the lack of adequate protection measures from external weather changes. 

The estimated values for each developmental stage in the linear regression model were as follows: y = 0.0034x − 0.0471 (χ2 = 3856.3; df = 3; *p* < 0.01) for the period from egg to adult; y = 0.0289x − 0.4536 (χ2 = 93.6; df = 3; *p* < 0.01) for the egg; y = 0.0070x − 0.0964 (χ2 = 875.2; df = 3; *p* < 0.01) for the larva; and y = 0.0088x − 0.1256 (χ2 = 546.9; df = 3; *p* < 0.01) for pupa (Figure 1, Table 2). The lower temperature threshold ranged from 10.0 °C to 15.7 °C. The thermal requirement in degree-days was calculated as 34.3 ± 3.1 for eggs, 142.8 ± 3.5 for larvae, 113.4 ± 4.2 for pupae, and 285.3 ± 7.5 for the whole growth period. To verify the accuracy of the derived linear regression analysis, the outbreak of adult insects was predicted in Chengju, Boeun, and Taean, Korea, which showed differences of +2–+4 days in Chengju, −1–+1 day in Boeun, and +4–+6 days in Taean (Table 3).

Table 4 presents the corn feeding amount of *S. frugiperda* according to temperature. The corn leaf intake of larvae bred was the highest at 16 °C (6.61 g) and lowest at 36 °C (2.9 g). In contrast, the daily feeding amount of *S. frugiperda* larvae was the highest at 28 °C and 32 °C and lowest at 16 °C. In some cases, however, the standard deviation of feeding amounts of the first and second instar larvae of *S. frugiperda* was equal to or higher than the average. Since the larvae’s feeding amount was so small in the first and second instars, this deviation was larger than the average deviation. In terms of the entire larval period, the feeding amount of first and second instar larvae was insignificant. Therefore, we concluded that feeding amounts can be determined fairly according to the temperature.

According to the seasonal occurrence trend of *S. frugiperda*, the outbreak of adult insects in 2020 and 2021 started in the third week of August and lasted until the third week of October (Figure 2). The peak season was late September; the number of outbreaks in Chungcheongnam-do was higher than that in Chungcheongbuk-do (Figure 3).

## 4. Discussion

Along with the global problem of climate change, the increase in the international trade of agricultural products and the development of transportation methods are accelerating the inflow and spread of foreign insects [18]. The agricultural sector is the most affected, particularly by the damage caused by new and unexpected pest occurrences. Farmers can reduce the density of existing pests by selecting an appropriate control method based on their knowledge. However, it is difficult to control pests at an early stage of an outbreak when pests with low density dramatically increase and cause damage or when identified pests suddenly infest. Pests such as *Lycorma delicatula*, *Metcalfa pruinosa*, *Hyphantria cunea*, and *Paratlanticus ussuriensis*, which were recently identified or showed a sudden increase in density in Korea, have now completely adapted to the harsh winter temperatures of Korea [19,20,21,22,23,24]. In addition, Lee et al. (2020) [14] reported the outbreak of *S. frugiperda* in Korea in 2019. Currently, *S. frugiperda* is naturally controlled in the winter owing to its inability to withstand the cold Korean winter temperatures.

According to the linear regression analysis results of the current study (Figure 1, Table 2), the R^2^ value of all development stages was 0.9 or more, suggesting linear growth of *S. frugiperda* under the breeding temperature of 20~32 °C. A previous study [25] reported that *S. frugiperda* develops linearly within a favorable temperature range. Hence, the temperature range of 20–32 °C is considered to be an ideal range for *S. frugiperda*’s growth. The lower temperature threshold was calculated as a minimum of 10 °C (second instar larva) and a maximum of 15.7 °C (egg; first instar larva), and thermal requirement in degree-days was calculated to be 14.4 °C. The minimum temperature from May to October 2021, which is the corn growing period in Korea [26], was 5.1 °C in May, 12.6 °C in June, 18.8 °C in July, 18.5 °C in August, 14.4 °C in September, and 1.4 °C in October. In Korea, *S. frugiperda* is expected to exhibit normal growth from July to September. In June, although the temperature is slightly lower, *S. frugiperda* is still expected to grow at the average minimum temperature in the southern region of Korea, which has a slightly higher temperature. The comparison of the calculated growth model with actual data using linear regression analysis showed a difference of +2–+4 days in Cheongju, −1–+1 days in Boeun, and +4–+6 days in Taean (Table 3). These differences were all within the standard deviation, making it possible to estimate the time of outbreaks at agricultural sites.

The growth period of *S. frugiperda* decreased by 47.6 days when the temperature increased from 16 °C to 20 °C and by 20.1 days when the temperature increased from 20 °C to 24 °C (Table 2). In other words, the number of growing days decreased by approximately 50% when the temperature increased from 16 °C to 24 °C, but no such decrease occurred above 24 °C. The growth of *S. frugiperda* was stabilized to some extent at temperatures of 24 °C and above. In addition, the survival rate of *S. frugiperda* larvae (Appendix A) was more than 97% at 24 °C to 32 °C, indicating that the temperature had no effect on survival. However, the survival rate dropped to 45.3% at 36 °C, which is estimated to be because *S. frugiperda* reached the upper temperature limit for growth. Therefore, the temperature range of 24 °C to 32 °C is suggested as the most suitable temperature for the growth of *S. frugiperda*. 

The results of the present study were compared with those of a study by Du Plessis et al. (2020) [27] on *S. frugiperda* growth period according to temperature (18–32 °C). Du Plessis et al [27]. reported a growth period of 71.4 days from the egg to the adult at the lowest temperature of 18 °C. However, in the present study, the growth period at this temperature was 97.2 days and 49.6 days at similar temperatures of 16 °C and 20 °C. The results of both the studies are comparable as 18 °C is an intermediate temperature between 16 °C and 20 °C and the growth period, i.e., 71.4 days, is roughly the average of 97.2 days and 49.6 days. However, the growth period from egg to adult at 32 °C was 16.1 days long in the current study, which is approximately 4 days shorter than the 20.3-day-long growth period at 32 °C in their study. Moreover, slight differences were observed at other temperatures, which is considered to be a characteristic of the local population. The samples in their study were collected from South Africa, whereas the samples in the present study were collected from Korea. In addition, we compared the growth period data from Barfield et al. (1978) [28], which were measured at a similar temperature to ours. According to their study, the growth period was 66.5 days at 21.1 °C, 37.8 days at 23.9 °C, and 19.3 days at 32.2 °C. We found that the growth period in our study (Table 1) was shorter than that of Barfield et al. (1978) [28], which was 49.6 days at 20 °C, 29.5 days at 24 °C, and 16.1 days at 32 °C. As the temperature decreased, the difference grew larger. Ali et al. (1990) [29] reported that cotton leaf larvae developed 37% slower than artificial feed larvae or corn leaf larvae when grown on cotton leaves. Unlike Barfield et al. (1978) [28], this study used corn leaves as feed instead of artificial feed, which may explain the difference in the growth period.

The highest amount of feeding (6.6 g) was observed at 16 °C and the lowest (0.9 g) was observed at 36 °C (Table 4). According to the feeding amounts, the damage seems to be most severe at 16 °C, which is natural because the growth period is the longest at 16 °C as the growth is reduced owing to low temperature. The daily corn leaf feeding amount of *S. frugiperda* was compared to determine the temperature at which feeding is most active. Damage was most severe at 28 °C and 32 °C owing to intensive harm for a short time period (Figure 4). The average summer temperature in the southern regions of South Korea (Gyeongnam, Jeonnam, and Jeju) is 22.3 °C in June, 26.4 °C in July, 26.0 °C in August, and 22.6 °C in September [26]. Therefore, the temperature from July to August was found to be most suitable for the growth of *S. frugiperda*. As the highest temperature of daytime is often recorded as an appropriate temperature for growth, cornfields should be frequently checked for *S. frugiperda* larvae from July to August. Moreover, additional attention is required because corn planted between July and August is young, and the corn yield may decrease by up to 70% if infected [5,6].

Luginbill (1928) [30] reported that the crabgrass intake ratio from the first to sixth instar larvae of *S. frugiperda* was 0.1, 0.6, 1.1, 4.7, 16.3, and 77.2, respectively. Calculated by area, they were 20.75, 82.25, 149, 644.5, 2,244.7, and 10,665 mm^2^. We calculated the food intake ratio using the same method used by Luginbill et al. (1928) [30], although the type of food was different, and no area was measured. The ratio in our study was somewhat similar when comparing the first to fourth instar larvae, but it was lower for the fifth and sixth instar larvae than that reported by Luginbill (1928) [30]. In contrast, when we calculated the amount of increase, both our study and Luginbill (1928) [30] found a relatively small increase for the first to fourth instar larvae, while exhibiting an explosive increase for the fifth and sixth instar larvae. This observation demonstrates how important it is to control *S. frugiperda* in its early stages.

To predict the timing of migration and control from 2020 to 2021, the occurrence pattern of adult *S. frugiperda* was analyzed using pheromone traps (Figure 2). In both 2020 and 2021, adults began to appear in the third week of August, and the incidence was insignificant until the second week of September, but then showed a tendency to increase significantly from the third week of September. Adults stopped being caught in the pheromone trap in the third week of October. During November, the average low temperature in the area where the pheromone trap was installed was 5 °C at Taean, 3 °C at Dangjin, 6.8 °C at Hongseong, −0.3 °C at Goesan in North Chungcheong Province, 1.7 °C at Cheongju, and 0.7 °C at Boeun [26]. Since the range of the lower temperature threshold for *S. frugiperda* is 10.0–15.7 °C according to our study, we speculate that they all died because they could not handle the cold temperature. The best period for *S. frugiperda* growth is when food plants are abundant due to abundant rainfall and moderate temperature [30]. August in Korea is hot, humid, and rainy, so *S. frugiperda* food crops are abundant during this time. Accordingly, we speculate that most of the larvae introduced from abroad in 2020–2021 survived in favorable weather conditions, resulting in a significant increase in adult occurrences in September.

The temperature data from the Korea Meteorological [26] were substituted into the *S. frugiperda* growth model used in this study. Based on the peak occurrence period of *S. frugiperda* inflow into Korea, it can be estimated that the Chungbuk region was intensively spawned between 14 August and 20 August, and the Chungnam region between 18 August and 24 August. The average temperature in Chungbuk and Chungnam during the growth period of *S. frugiperda* from an egg to an adult, which is from mid-August to mid-September, was 21.7 °C and 22.3 °C, respectively. These temperatures do not interfere with its growth; hence, it easily spreads in Korea.

Considering the growth period by temperature, feeding amount, daily feeding amount, and survival rate of *S. frugiperda*, the most suitable temperature for growth was determined to be from 28 °C to 32 °C, which is similar to the summer temperature in Korea [26]. As of 2021, there were 122,044 farms and 16,185 ha of area under corn cultivation in Korea [31]. In Korea, corn is intensively cultivated in the summer season [32]. Particular attention is required during this season as this time overlaps with the optimal temperature of *S. frugiperda* growth. 

The best way to prevent the spread of *S. frugiperda*-induced damage is to control it at an early stage of the outbreak. To obtain control in the early stage of the outbreak, the accurate prediction of the expected time of an outbreak is essential. For common pests, outbreak timing is predicted using the winter and spring temperature of wintering individuals; however, for *S. frugiperda*, which migrates from abroad [14], it should be predicted based on the detection time of the migrating population. Therefore, it is possible to obtain control at an early stage using the growth and development model in this study along with an understanding of the migrating population using pheromone traps. Furthermore, as corn grows to a height taller than that of humans, it is difficult for human workers to directly enter the corn field and control the outbreak; therefore, the use of drones is necessary in this case. 

In conclusion, as *S. frugiperda* outbreaks should be predicted based on the migrating population, it is crucial to understand the exact migration timing; however, it is impossible for a few researchers to cover large areas. Therefore, a collaboration between various organizations is of utmost importance. Furthermore, corn-growing farmers must frequently monitor *S. frugiperda* and report their sightings to relevant organizations. 

## Figures and Tables

**Figure 1 insects-13-00897-f001:**
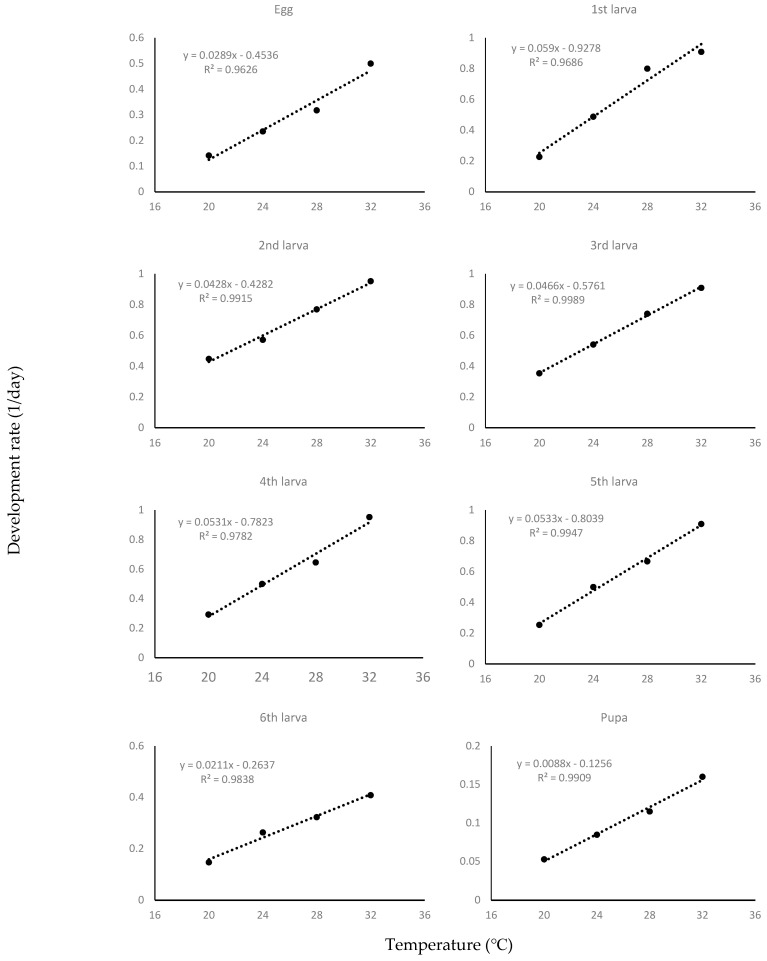
Liner regression analysis of temperature and growth of *Spodoptera frugiperda* (from egg to pupa).

**Figure 2 insects-13-00897-f002:**
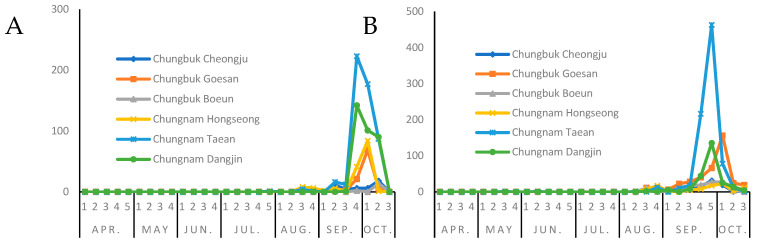
Seasonal occurrence trends of adult *Spodoptera frugiperda* from 2020 to 2021 (**A**): 2020; (**B**): 2021).

**Figure 3 insects-13-00897-f003:**
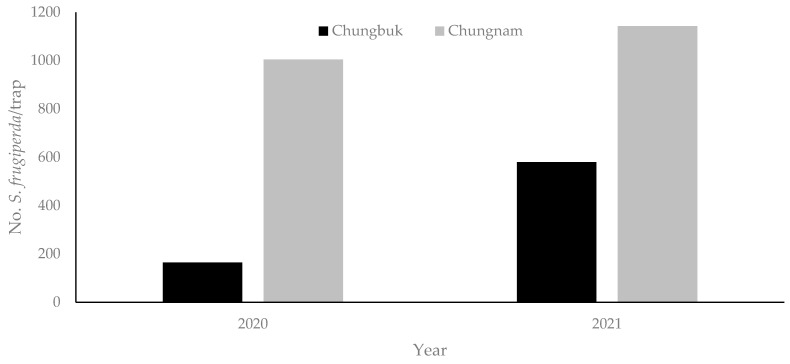
Comparison of adult *Spodoptera frugiperda* occurrence in Chungcheongbuk-do (Chungbuk) and Chungcheongnam-do (Chungnam) from 2020 to 2021.

**Figure 4 insects-13-00897-f004:**
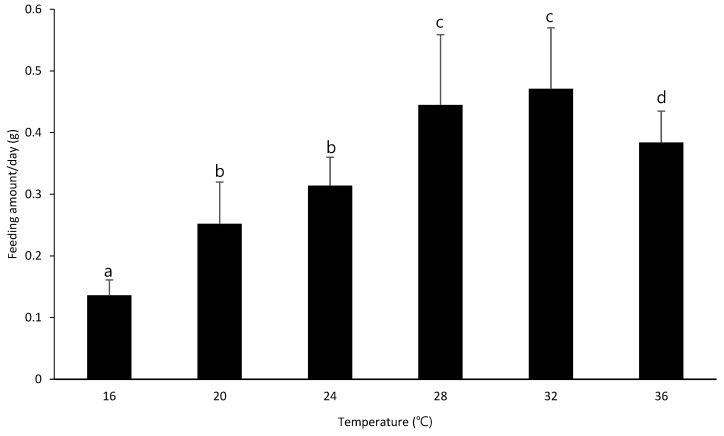
Daily corn leaf feeding amount of *Spodoptera frugiperda* according to temperature (The same letters are not significantly differenct (*p* > 0.05) by Tukey’s HSD test).

**Table 1 insects-13-00897-t001:** Development time (days ± SD) of *Spodoptera frugiperda* according to temperature (day).

Temp. (°C)	Development Time (Day)								
	Eggs	Larvae Stage						Pupae	Total
		1st	2nd	3rd	4th	5th	6th		
16	10.1 ± 0.7 a	6.4 ± 0.7 a	4.1 ± 0.6 a	6.5 ± 0.6 a	6.5 ± 0.5 a	11.3 ± 0.8 a	16.7 ± 0.7 a	35.3 ± 1.1 a	97.2 ± 1.2 a
20	7.1 ± 0.2 b	4.4 ± 0.7 b	2.2 ± 0.4 b	2.8 ± 0.7 b	3.4 ± 0.5 b	3.9 ± 0.7 b	6.8 ± 0.5 b	18.9 ± 0.8 b	49.6 ± 1.2 b
24	4.3 ± 0.7 c	2.1 ± 0.7 c	1.8 ± 0.4 c	1.9 ± 0.4 c	2.0 ± 0.4 c	2.0 ± 0.4 c	3.8 ± 0.9 c	11.8 ± 0.6 c	29.5 ± 1.4 c
28	3.2 ± 0.5 d	1.3 ± 0.4 d	1.3 ± 0.5 d	1.4 ± 0.5 d	1.6 ± 0.6 cd	1.5 ± 0.6 cd	3.1 ± 0.4 cd	8.7 ± 1.0 d	21.9 ± 1.7 d
32	2.0 ± 0.3 e	1.1 ± 0.3 d	1.1 ± 0.2 d	1.1 ± 0.3 d	1.1 ± 0.2 d	1.1 ± 0.3 d	2.5 ± 0.5 de	6.3 ± 0.5 e	16.1 ± 1.1 e
36	2.0 ± 0.2 e	1.0 ± 0.0 d	1.0 ± 0.0 d	1.0 ± 0.0 d	1.3 ± 0.7 d	1.6 ± 0.8 cd	2.4 ± 0.6 e	5.9 ± 0.7 e	15.5 ± 0.7 e

Means within a row followed by the same letter are not significantly different (*p* < 0.05) by Tukey’s HSD test.

**Table 2 insects-13-00897-t002:** Linear regression analysis, lower temperature threshold (LTT), and thermal requirement in degree-days (TRD) during the developmental stages of *Spodoptera frugiperda* according to temperature.

Development Stage	Regression Model	R^2^ Value	LTT (°C)	TRD
Eggs	y = 0.0289x − 0.4536	0.9626	15.7	34.3 ± 3.1
1st instar	y = 0.0590x − 0.9278	0.9686	15.7	17.3 ± 1.3
2nd instar	y = 0.0428x − 0.4282	0.9915	10.0	23.3 ± 0.8
3rd instar	y = 0.0466x − 0.5761	0.9989	12.4	21.5 ± 0.2
4th instar	y = 0.0531x − 0.7823	0.9782	14.7	18.8 ± 1.0
5th instar	y = 0.0533x − 0.8039	0.9947	15.1	18.8 ± 0.6
6th instar	y = 0.0211x − 0.2637	0.9838	12.5	47.7 ± 2.7
larvae	y = 0.0070x − 0.0964	0.9977	13.8	142.8 ± 3.5
Pupae	y = 0.0088x − 0.1256	0.9909	14.3	113.4 ± 4.2
Egg to adult	y = 0.0034x − 0.0490	0.9958	14.4	285.3 ± 7.5

**Table 3 insects-13-00897-t003:** Comparison of the calculated and actual data of the adult *Spodoptera frugiperda* outbreak.

Location	Estimated Date	Number of Eclosions (Observed Days)			Deviation (Day)
Cheongju	22 September	12 (24 September)	7 (25 September)	6 (26 September)	+2–+4
Boeun	28 September	17 (27 September)	6 (28 September)	3 (29 September)	−1–+1
Taean	16 September	10 (20 September)	9 (21 September)	2 (22 September)	+4–+6

**Table 4 insects-13-00897-t004:** Corn leaf feeding amount (mean ± SD) by the development stages of *Spodoptera frugiperda* according to temperature.

Larvae Stage	Feeding Amount (g)					
	16 °C	20 °C	24 °C	28 °C	32 °C	36 °C
1st	0.029 ± 0.029 a	0.026 ± 0.029 ab	0.008 ± 0.007 ab	0.012 ± 0.015 ab	0.006 ± 0.005 b	0.027 ± 0.033 ab
2nd	0.040 ± 0.034 a	0.026 ± 0.028 ab	0.028 ± 0.025 ab	0.010 ± 0.010 b	0.009 ± 0.004 b	0.035 ± 0.035 a
3rd	0.164 ± 0.106 a	0.100 ± 0.063 ab	0.088 ± 0.084 b	0.104 ± 0.077 ab	0.065 ± 0.059 b	0.068 ± 0.039 b
4th	0.372 ± 0.181 a	0.386 ± 0.154 a	0.364 ± 0.182 a	0.372 ± 0.273 a	0.255 ± 0.117 a	0.239 ± 0.106 a
5th	1.454 ± 0.335 a	1.172 ± 0.442 ab	0.867 ± 0.385 bc	0.538 ± 0.237 c	0.567 ± 0.437 c	0.495 ± 0.275 c
6th	4.526 ± 0.746 a	4.065 ± 0.990 a	2.452 ± 0.593 c	3.640 ± 1.093 ab	2.716 ± 0.869 bc	2.255 ± 0.403 c
1st to 6th	6.610 ± 0.973 a	5.776 ± 0.829 a	3.772 ± 0.495 bc	4.678 ± 1.195 b	3.617 ± 0.625 c	2.896 ± 0.449 c

Means within a row followed by the same letter are not significantly different (*p* < 0.05) by Tukey’s HSD test.

## Data Availability

The data presented in this study are available on request from the corresponding author.

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
