# Peer review of "Prediction of Growth Characteristics and Migration Period of *Spodoptera frugiperda* (Lepidoptera: Noctuidae) According to Temperature"

_insects, 2022, doi:10.3390/insects13100897_

Round 1
Reviewer 1 Report
Dear Authors:
The study is exciting and has a lot of potential regarding S. frugiperda management. The results have a lot of scientific meaning and importance. However, I am missing some of the key elements of scientific study. For instance, there should be a clear statement regarding why the study needs to be done. Also, in the result section, I found seasonal monitoring of S. frugiperda but you should mention it in the introduction section. Similarly, the discussion section also needs to be improved to show the applied meaning of the results.
Some of the specific comments are as follow:
Line 8: An outbreak or first detected?
Line 44-50: Missing rationale for the study.
Line 53: 'The samples' : are you indicating larvae?
Line 65-66: Please rewrite the sentence
Line 79: Use separate format for the title or subtitle
Line 80-81: Please rewrite
Line 91-92: Please rewrite
Line 93-100: New topic? Please provide some background in introduction section
Line 107: '1/day' Please use standard format
Line 176-273: Please restructure the discussion section. For instance, line 191-192, should be moved in an appropriate section.
Author Response
Dear Reviewer
We appreciate your review of our paper very much. We strongly agree with your points and we have made efforts to address each of your comments. The manuscript has also been revised accordingly. Details of the changes made are given below.
Line 8: An outbreak or first detected?
Response: Thank you; ‘first detected’ is correct. We have modified the text accordingly.
Line 44-50: Missing rationale for the study.
Response: We apologize for this. We have added the rationale for the study in the Introduction section.
Line 53: 'The samples': are you indicating larvae?
Response: Yes, we have corrected this in the revised manuscript.
Line 65-66: Please rewrite the sentence
Response: We have rephrased the sentence.
Line 79: Use separate format for the title or subtitle
Response: We have changed the font and numbered each line.
Line 80-81: Please rewrite
Response: We have rewritten the sentence.
Line 91-92: Please rewrite
Response: We have rewritten the sentence.
Line 93-100: New topic? Please provide some background in introduction section
Response: We have added more background to the Introduction section.
Line 107: '1/day' Please use standard format
Response: We referred to a number of additional papers and have modified the term to ‘1/day’
Line 176-273: Please restructure the discussion section. For instance, line 191-192, should be moved in an appropriate section.
Response: Thank you for your suggestion. The entire Discussion section has been reorganized and supplemented with additional references.

Reviewer 2 Report
The reviewer was surprised that the authors had not referenced previous early studies of this insect in areas where it has long been prevalent. Similar information about growth rates was published previously in the1920’s. A later paper by Sparks, “A review of the biology of the fall armyworm,” Fla. Entomol. 62: 82-87 (1979), summarizes the original work. The authors do refer to recent similar studies in South Africa (which also fail to refer to the original studies).
For this reason, the reviewer suggests that the manuscript include references to the previous growth rate literature and focus more strongly on the new information in the study. For example, the Korean populations have different growth patterns than the South African populations, and the information on seasonal occurrences in 2020 and 2021 (Figure 6) also is new. Possibly, Figs. 1-2 should be moved to an appendix, as they present information similar to earlier literature.
Author Response
The reviewer was surprised that the authors had not referenced previous early studies of this insect in areas where it has long been prevalent. Similar information about growth rates was published previously in the1920’s. A later paper by Sparks, “A review of the biology of the fall armyworm,” Fla. Entomol. 62: 82-87 (1979), summarizes the original work. The authors do refer to recent similar studies in South Africa (which also fail to refer to the original studies).
For this reason, the reviewer suggests that the manuscript include references to the previous growth rate literature and focus more strongly on the new information in the study. For example, the Korean populations have different growth patterns than the South African populations, and the information on seasonal occurrences in 2020 and 2021 (Figure 6) also is new. Possibly, Figs. 1-2 should be moved to an appendix, as they present information similar to earlier literature.
Response: We appreciate your review of our paper very much. We strongly agree with your point and accordingly, the Introduction and Discussion sections have been further supplemented with additional references. Furthermore, as per your suggestion, Figures 1 and 2 have been moved to the Appendix.
Round 2
Reviewer 1 Report
In line 8: 'outbreak' should be changed.